# Perceived Disaster Preparedness and Willingness to Respond among Emergency Nurses in South Korea: A Cross-Sectional Study

**DOI:** 10.3390/ijerph191811812

**Published:** 2022-09-19

**Authors:** Won-Seok Choi, Sung Youl Hyun, Hyunjin Oh

**Affiliations:** 1National Emergency Medical Center, National Medical Center, Seoul 03080, Korea; 2Department of Traumatology, College of Medicine, Gachon University, Incheon 21936, Korea; 3College of Nursing, Gachon University, Incheon 21936, Korea

**Keywords:** disaster preparedness, disaster response, willingness to respond, emergency nurse

## Abstract

Introduction: Emergency nurses serve a vital role in disaster situations. Understanding their disaster preparedness and willingness to respond to a disaster is important in maintaining appropriate disaster management. The purpose of this study was to explore emergency nurses’ disaster preparedness and willingness to respond based on demographic and disaster-related characteristics, and their willingness to respond based on specific disaster situations. Methods: In this descriptive, cross-sectional study, the Disaster Preparedness Questionnaire for Nurses and willingness to report to duty by type of event were used to collect data from 158 nurses working in four regional emergency medical centers from 1 December 2019 to 30 April 2020 in the early stages of the COVID-19 pandemic. Results: Emergency nurses with personal disaster experience as a victim or witness (*t* = 3.65, *p* < 0.001), professional disaster experience (i.e., working as a nurse) (*t* = 3.58, *p* < 0.001), who were current members of Korean Disaster Medical Assistance Teams (*t* = 6.26, *p* < 0.001), and who received disaster-related training within a year (*t* = 5.84, *p* < 0.001) showed a high level of perceived disaster preparedness. Emergency nurses who have professional disaster experience (i.e., working as a nurse) (*t* = 2.42, *p* = 0.017), are on a current disaster team (*t* = 2.39, *p* = 0.018), and have received disaster training (*t* = 2.73, *p* = 0.007) showed a high level of willingness to respond. Our study showed a high willingness to respond to natural disasters and low willingness to respond to technological disasters. Discussion: To promote the engagement of emergency nurses in disaster response, disaster education programs should be expanded. Enhancing the safety of disaster response environments through supplementing medical personnel, distributing available resources, and providing sufficient compensation for emergency nurses is also essential.

## 1. Introduction Background

A disaster is defined as “a situation or event, which overwhelms local capacity, necessitating a request to a national or international level for external assistance; an unforeseen and often sudden event that causes great damage, destruction, and human suffering” by the Center for Research on the Epidemiology of Disasters [1]. Disaster preparedness refers to all prior action plans and efforts implemented to establish a disaster response system before a disaster occurs [2]. To be prepared, emergency nurses need to have sufficient knowledge and skills to minimize the negative effects of a disaster including trauma, infectious diseases, and physical and psychological distress [2,3,4,5].

Willingness to respond (WTR) in a disaster means that healthcare workers respond to and approve a request to report to work in the event of a disaster [6]. In South Korea, institutions or staffs are required to work in the event of a disaster and approve such requests unless there are special reasons not to do so [7]. In the literature, the absenteeism of emergency responders during disasters is considered a critical issue that can lead to negative patient outcomes. The main reasons for role abandonment are the inability or unwillingness to report to work [6]. During the COVID-19 pandemic, healthcare providers experienced various difficulties such as heavy workloads, fear of infection, lifestyle changes, and psychological and physical struggles [8,9], and there were increasing concerns regarding hospital workforce shortages. In a recent study, about 48.2% of emergency nurses reported their intention to leave within 5 years after the first year of COVID-19 pandemic [10]. In South Korea, there was also a strike by unionized health workers demanding better working conditions and expansion of public health infrastructure during the pandemic [11].

Regional Emergency Medical Centers are designated medical institutions for emergency medical support in the event of a major disaster and provide education and practical training for emergency medical workers, including emergency nurses, in South Korea [7,12]. The Regional Emergency Medical Centers organize and manage the Korean Disaster Medical Assistance Teams (KDMAT), which consist of four people per team (one physician, two nurses or emergency medical technicians, and one administrative assistant) [12]. Emergency nurses serve a vital role in the emergency department and on KMATs in caring for victims, infection control, contingency planning to prevent further damage, triage, mass immunizations, mass evacuations, and the management of mass casualties during a disaster [13]. Nurses specialized in emergency, trauma, and disaster care must be prepared for disasters to optimize disaster response and recovery efforts and patient outcomes during disasters. In this regard, the literature shows that nurses are generally ill-prepared in their abilities to respond to disaster events [2,14,15].

The WTR of healthcare workers depends on the type of disaster [6], family and personal disaster plans, and car responsibilities [16]. Published literature shows that healthcare workers are reluctant to respond to so-called “dirty” or technological disasters such as a bomb explosions, chemical attacks, bioterrorism, pandemics, and general disasters compared to natural, environmental, or weather-related disasters [6]. In addition, the most common barriers to WTR were care responsibilities such as caring for children or pets [16]. In contrast, an older age and sufficient personal protective equipment (PPE) were significant facilitators of WTR [17].

Understanding emergency nurses’ disaster preparedness and WTR during a disaster can provide governments and public health leaders with vital information relevant to workforce implications and possible strategies to maintain sufficient and appropriate disaster management. A recent integrative review addresses the variations among the nations, as acceptable levels of perceived disaster preparedness have been reported in some developed countries [18]. However, little is known about perceived levels of disaster preparedness and the WTR of emergency nurses in South Korea. Thus, the aim of the current study was to explore emergency nurses’ perceived disaster preparedness and WTR based on demographic and disaster-related characteristics, and their WTR according to specific disaster situations.

## 2. Methods

### 2.1. Design

This study used a cross-sectional design and employed a self-reported questionnaire.

### 2.2. Study Participants and Collection

The study participants consisted of emergency nurses working in four regional medical institutions in South Korea (tertiary hospitals located in Incheon, Gyeonggi, Daejeon, and Chonnam regions). Participants were provided with an informed written consent form to participate therein. A priori computation of the sample size using G* Power version 3.1 revealed that 147 participants were required for *t*-tests with an effect size (f) of 0.3, an alpha value of 0.05, and an actual power of 0.95.

Data were collected from 1 December 2019 to 30 April 2020. Permission to collect data was obtained from the directors or emergency department managers in the four hospitals. Permission for this study was obtained from the Gachon University Institutional Review Board (no. 1044396-201908-HR-142-02). In total, 160 emergency nurses between four hospitals were invited to participate in this study, and 158 completed a paper questionnaire and returned the survey. The response rate was about 98%.

### 2.3. Instrument

#### 2.3.1. General and Disaster-Related Characteristics

The general characteristics of participants [19] included gender [20], age [17], education, religion, marital status [21], family, and having a pet to care for [19]. Disaster-related characteristics included personal disaster experience, professional disaster experience (i.e., working as a nurse in a disaster), KDMAT status, disaster-related education experience, safety concerns (self, family, others), and available resources in the case of a disaster.

#### 2.3.2. Disaster Preparedness

Nurses’ disaster preparedness refers to the degree of preparation that can minimize the negative impact and consequences of a disaster and includes the needed knowledge and skills [3]. Disaster preparedness was assessed using the Disaster Preparedness Questionnaire for Nurses (DPQ-N) developed by Ann et al. [2]. It consists of 63 items with 11 sub-categories. Each item was scored on a five-point Likert scale ranging from 1 (strongly disagree) to 5 (strongly agree).

The 10 DPQ-N domains were reduced to 9 distinctive domains and showed stable psychometric properties (CVI = 0.88, cumulative variance explained = 71.3%, Cronbach’s α = 0.86~0.94). The subjects’ disaster preparedness was 2.79 (SD 0.55).

The total scores were calculated by summing the item scores and ranged from 63 to 315. Higher scores indicate a higher level of preparedness associated with disasters. In this study, internal reliability was 0.98.

#### 2.3.3. Willingness to Respond (WTR)

To investigate WTR, we adopted the willingness to report to work during different types of catastrophic events from Qureshi et al. (Catastrophic disaster scenarios used for facilities in and around New York City) [22]. The original question consisted of seven scenarios for heavy snow, smallpox, chemical terrorism, explosive terrorism, asthma caused by forest fires, radioactive bombing, and SARS. In our study, in accordance with local conditions, eight scenarios were asked about, namely heavy snow, heavy floods, chemical plant fires (toxic substances), explosive terrorism (MCI explosion), earthquakes (building collapse), radioactive leaks, Middle East Respiratory Syndrome (MERS), and war (local provocation with North Korea). To assess WTR, participants were asked to respond to each scenario with 1 point for “yes” and 0 for “no willingness to respond” and “not sure”.

### 2.4. Data Analysis

Data were analyzed descriptively using IBM SPSS software (version 25.0; IBM Corp., Armonk, NY, USA). All study variables were screened for suspected errors, missing data, and outliers, and the questionnaires with missing data were excluded from analysis. Normality test was done using histograms and the Shapiro–Wilk test. Study variables were summarized as frequencies and percentages for categorical variables and the mean and standard deviation (SD) for continuous variables. *T*-tests and ANOVAs were performed to compare the differences in study variables by disaster preparedness and WTR. The level of significance was set at *p* < 0.05.

## 3. Results

The emergency nurses who participated in this study from the Regional Emergency Medical Centers were aged between 22 and 50 years, with an average age of 29 years. The majority of nurses were female, although the proportion of male nurses was similar to that of the national average. Figure 1 provides the findings regarding emergency nurses’ perceived level of disaster preparedness with respect to each competency. Participants’ mean score on the DPQ-N was 3.33 (SD 0.71) out of 5, ranging from 2.54 to 3.94 (Figure 1). Among the 11 subcategories, emergency patient care (M 3.94, SD 0.75) scored the highest, followed by personal preparedness (M 3.53, SD 0.91), psychological issues (M 3.47, SD 0.87), legal and ethical issues (M 3.39, SD 0.85), hospital disaster planning (M 3.24, SD 0.98), epidemiology and quarantine (M 3.13, SD 0.95), disaster resources (M 3.12, SD 0.80), overall preparedness (M 3.07, SD 1.04), communication (M 3.05, SD 1.03), basic concept of disaster (M 3.03, SD 0.92), and Chemical, Biological, Radiological, Nuclear, and high yield Explosives (CBRNE) agents (M 2.54, SD 1.01).

### 3.1. Differences in Disaster Preparedness and WTR by Demographic Characteristics

Table 1 presents the differences in disaster preparedness and WTR according to participants’ demographic characteristics. The results indicate that educational status (college graduate) (F = 4.09, *p* = 0.018), marital status (*t* = 2.42, *p* = 0.017), having children (*t* = 2.64, *p* = 0.009), and having more years of clinical experience (*t* = −2.43, *p* = 0.016) were associated with increased perceived level of disaster preparedness. Although there was no statistical difference, participants who were female, had a higher level of education, identified as religious, were single, had children, had dependents excluding children, had pets, had a support system, had longer clinical experience, and had a higher job position reported higher levels of WTR.

### 3.2. Differences in Disaster Preparedness and WTR by Disaster-Related Characteristics

Table 2 presents the differences in disaster preparedness and WTR according to participants’ disaster-related characteristics. Nurses with personal disaster experience (M 3.68, SD 0.75), professional disaster experience (M 3.83, SD 0.68), on a disaster team (M 3.81, SD 0.63), and who received disaster training within a year (M 3.59, SD 0.62) reported higher levels of perceived disaster preparedness. The average WTR was higher among participants with professional disaster experience (i.e., working as a nurse), (M 5.32, SD 2.48), who were current members of a disaster team (M 5.02, SD 2.54), and in those with recent disaster training (M 4.76, SD 2.71). Nearly 80% of the participants answered that resources would not be sufficiently supplied and there would be inadequate compensation (including lack of worker’s compensation insurance) when participating in a disaster event.

### 3.3. WTR by Disaster Event

Figure 2 graphically depicts the distribution of WTR by disaster event type. Emergency nurses were more willing to respond to snow (70%), earthquake (68%), floods (65%), and wars (55%), and less willing to respond to a radiation (21%), chemical events (43%), and MCI explosion (47%) situations.

## 4. Discussion

Emergency nurses in Regional Emergency Medical Centers serve an important role in a disaster event. Our study explored the differences in disaster preparedness and WTR by demographic and disaster-related characteristics, and WTR by disaster event of emergency nurses working in Regional Emergency Medical Centers in South Korea.

In our study, emergency nurses who graduated from 3-year college (F = 4.09, *p* = 0.018), are married (*t* = 2.42, *p* = 0.017), have children (*t* = 2.64, *p* = 0.009), and have more years of clinical experience (*t* = −2.43, *p* = 0.016) showed higher levels of perceived disaster preparedness. Nurses who graduated from 3-year college were more familiar with these issues than those with a higher degree. According to a previous study from South Korea [23], there was no significant difference between the level of preparedness competence and educational status. This could be because about 95% of the nursing universities in South Korea have recently included disaster-related education in the undergraduate curriculum [24]. Emergency nurses who are married, have children or family caregiving responsibilities may be more interested in their safety and security; thus, the level of personal disaster preparedness would positively affect their disaster preparedness competence.

In our study, the differences in WTR according to personal characteristics were not statistically significant. However, those with a higher education level, religion, single status, no children, no pets, and more clinical experience showed a higher level of WTR. Literature on the willingness to work in a disaster showed that the responsibility and concerns regarding caring for family (children and the elderly), a lack of adequate means of transportation, and individual safety issues were important factors [15,25]. Our study results are consistent with a recent Korean study in which no difference was found in WTR depending on the presence or absence of children [26]. The participant characteristic that most nurses (84.3%) in our study did not have children might contribute to these consistent results.

Emergency nurses with personal disaster experience as a victim or witness (*t* = 3.65, *p* < 0.001), professional disaster experience (i.e., working as a nurse) (*t* = 3.58, *p* < 0.001), who were current members of KDMAT (*t* = 6.26, *p* < 0.001), and who received disaster-related training within a year (*t* = 5.84, *p* < 0.001) showed a high level of perceived disaster preparedness which is consistent with the literature [3,5,13]. This also implies that continued education opportunities (e.g., mock disaster drills) are critical for emergency nurses’ disaster preparedness. In addition, there was a statistically significant difference between nurses who received disaster training within a year and those who did not. Thus, continued education and training programs are essential in preparing nurses for these events [27,28].

Emergency nurses who have professional disaster experience (i.e., working as a nurse) (*t* = 2.42, *p* = 0.017), are on a current disaster team (*t* = 2.39, *p* = 0.018), and have received disaster training (*t* = 2.73, *p* = 0.007) showed a high level of WTR in our study. Healthcare professionals who had both disaster training and disaster experience were more likely to report willingness to respond to disaster events [15,19]. Our results are consistent with those of previous studies showing that nurses with disaster training had a 1.5 times increased likelihood of WTR [29] and those having skills and/or knowledge in disasters were four times more likely to demonstrate WTR during a disaster [16]. Although the participants in our study were emergency nurses in a Regional Emergency Medical Center, only approximately 58% had received disaster-related education. In a recent Korean study, most nurses (86.2%) in two public hospitals had received some form of disaster education [26]. Nurses in reginal hospitals in South Korea might have more opportunities for disaster education. Currently, there are available education programs such as disaster nursing education at the Armed Forces Nursing Academy, Disaster Relief Training at the Center for Disaster Training and Research of Yonsei Severance Hospital, and Korean Disaster Life Support (KDLS) by National Emergency Medical center in South Korea. KDLS is a curriculum currently provided only to the KDMATs of Regional Emergency Medical Centers, who provide simulation education programs for each disaster scenario to about 50 people 4 times a year for free. To promote the engagement of public health and the response to disasters for emergency nurses, disaster-related education centers and educational opportunities on various types of disasters should be expanded and implemented.

Although there were no statistically significant differences, the rates of concern for family (85.5%) and for self (30.8%) were consistent with those found in a previous study [6]. In previous studies, safety concerns for the family or self were presented as obstacles to WTR in disasters [8,16,30]. In addition to family safety, the ability to communicate with family members while away from home is an important concern for healthcare workers [30]. Arbon et al. showed that 72.9% of participants had no plan for disaster preparation for themselves and their families, and those with disaster plans for their families demonstrated 8 times higher WTR [16].

Nearly 52% of the participants in our study answered that resources would not be sufficiently supplied (F = 2.10, *p* = 0.0129) and 50.6% said that there would be inadequate compensation when participating in a disaster event (F = 2.13, *p* = 0.012). This suggests emergency nurses may have low levels of trust in their institutions and governments. PPE is an essential factor in the safety of disaster response participants. A Western study reported that almost all study participants (94%) were confident that institutions would supply sufficient PPE for their safety, and that the capability to supply resources affect WTR during a disaster [17]. Nurses who participated in providing patient care for MERS in South Korea reported anxiety and the burden of the risk of transmission of new infectious diseases, as well as an unprepared treatment environment that included a lack of resources [31,32]. Inadequate compensation, such as not enough incentive or hazard pay, is an important barrier to WTR [15,25]. In addition, dependent caregiving services and emergency transportation plans for staff should be incorporated in Emergency Operation Plans of the Regional Emergency Medical Centers [22]. The data of the current study were collected in 2019, just before and during the early stages of the COVID-19 pandemic. However, problems arose because sufficient PPE and compensation were not given to healthcare providers during COVID-19 [9]. Currently, Korea’s COVID-19 quarantine is considered one of the best practices worldwide, but nurses are still working in an unsafe environment because of an insufficient number of public hospitals and a lack of nurses. To improve WTR, it is essential to create a safe environment such by supplementing medical personnel, obtaining and distributing available resources, and providing sufficient compensation for healthcare providers.

Our study confirmed a high WTR to natural disasters such as snow (69.8%), earthquakes (67.9%), floods (65.4%) and MERS (53.5%), and a low WTR to technological disasters such as radiation (21.4%), chemical events (43.4%), and MCI explosions (47.2%). These results are consistent with those of previous studies [15,16] finding a higher WTR to natural disasters (e.g., floods and earthquakes) than technological disasters (e.g., radioactive events). The ambiguity and uncertain nature of technological disasters may cause fear and have a negative effect, decreasing nurses’ WTR [15,32]. The fact that most participants (80%) showed concerns that sufficient resources would not be supplied may be related to the low WTR to technological disasters in our study. The relationship between safety concerns and WTR during epidemics of unknown contagious diseases has been reported [15].

The findings of our study cannot be generalized to all emergency nurses because this research was conducted at four Regional Emergency Medical Centers with a population recruited through convenience sampling. Considering the sample characteristics of emergency nurses from Regional Emergency Medical Centers, the concept of disaster preparedness may be more infused into the culture, and training opportunities are more readily available compared to other facilities. In addition, our study explored emergency nurses’ preparedness through the DPQ-N in a self-reporting manner. Thus, readers should pay attention to the interpretation of these self-reported results.

## 5. Implications for Emergency Clinical Care

Clinical practice implications are based on the differences in disaster preparedness and WTR according to emergency nurses’ demographic and disaster-related characteristics. Emergency nurses with personal or professional disaster experience, experience on a disaster team, and who received disaster training within a year reported high perceived levels of disaster preparedness. Nearly 80% of emergency nurses reported anticipating that resources would not be sufficiently supplied and there would be inadequate compensation, including hazard pay and worker’s compensation insurance, when participating in a disaster event. Our evidence gathered before and during the early pandemic period may be helpful for a better understanding of the perception and readiness of frontline emergency nurses, and ways to enhance emergency nurses’ willingness to respond for future disaster response. Hospital administrators and policymakers should consider the facilitators and systematically address the barriers to willingness to respond and provide appropriate disaster-specific education and training.

## 6. Conclusions

Our study is the first in South Korea to explore emergency nurses’ characteristics and differences in disaster preparedness and WTR. This research highlighted that WTR differs depending on having professional disaster experience (i.e., working as a nurse), being a member of the KDMAT, and having received disaster training within a year. Only about 20% of emergency nurses expressed a positive expectation regarding having sufficient available resources and receiving compensation for disaster participation.

We suggest that disaster-related education centers, educational opportunities for various disaster education programs, and budgets of training be expanded in South Korea. In addition, it is essential to create a safe environment such as by supplementing medical personnel, distributing available resources, providing sufficient compensation (incentives, hazard pay, and wide coverage of compensation insurance) for emergency nurses. Finally, increasing nurses’ perception of safety and security during a disaster is important in increasing WTR.

## Figures and Tables

**Figure 1 ijerph-19-11812-f001:**
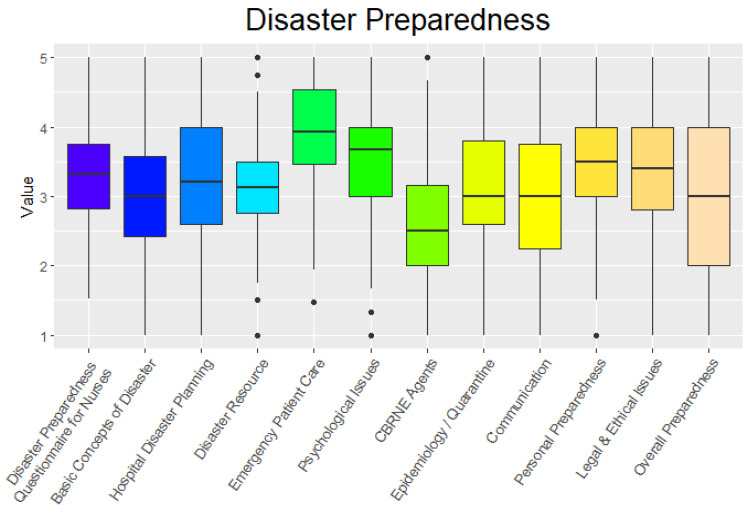
Results of the Disaster Preparedness Questionnaire for Nurses. CBRNE: Chemical, Biological, Radiological, Nuclear, and high yield Explosives.

**Figure 2 ijerph-19-11812-f002:**
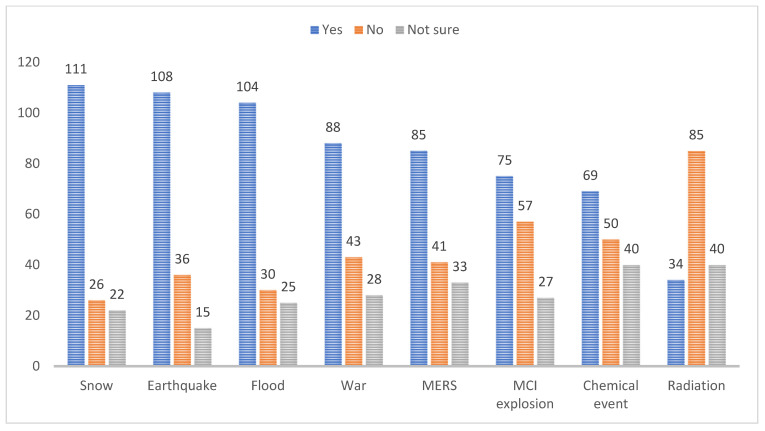
Willingness to respond by disaster events.

**Table 1 ijerph-19-11812-t001:** Differences in disaster preparedness and WTR by demographic characteristics (*N* = 158).

Characteristics	Categories	*n*	%	DPQ-N		*t*/F(*p*)	WTR		*t*/F(*p*)
				Mean	SD		Mean	SD	
Gender	Female	130	82.3	3.30	0.69	1.32(0.19)	4.28	2.64	0.11(0.911)
	Male	28	17.7	3.49	0.81		4.21	3.02	
Age		29.27 (5.37) ^†^		3.33	0.71		4.27	2.70	
Educational status	3-year College graduate	25	15.8	3.64	0.58	4.09(0.018)	4.4	3.27	0.20(0.817)
	Bachelor’s	111	70.3	3.23	0.72		4.18	2.59	
	Master’s or higher	22	13.9	3.49	0.75		4.55	2.61	
Religion	Yes	56	35.4	3.41	0.75	1.01(0.315)	4.57	2.78	1.06(.293)
	No	102	64.6	3.29	0.70		4.10	2.65	
Marital status	Married	43	27.2	3.55	0.67	2.42(0.017)	3.91	2.92	−1.02(0.308)
	Single	115	72.8	3.25	0.71		4.4	2.61	
Children	Yes	25	15.8	3.67	0.61	2.64(0.009)	3.84	3.00	−0.86(0.391)
	No	133	84.2	3.27	0.72		4.35	2.64	
Dependents	Yes	61	38.6	3.46	0.79	1.79(0.076)	4.62	2.68	1.32(0.188)
(Excluding children)	No	97	61.4	3.25	0.66		4.04	2.70	
Pets	Yes	35	22.2	3.14	0.58	−1.76(0.079)	4.46	2.65	0.47(0.636)
	No	123	77.8	3.28	0.74		4.21	2.72	
Support system	Yes	83	52.5	3.34	0.68	0.18(0.858)	4.45	2.53	0.88(0.379)
	No	75	47.5	3.32	0.75		4.07	2.87	
Perceived health problems	Yes	7	4.4	3.62	0.97	0.18(0.858)	5.29	2.29	1.02(0.308)
	No	151	95.6	3.32	0.70		4.22	2.71	
Clinical experience (year)	Short (<5)	80	50.6	3.20	0.69	−2.43(0.016)	4.36	2.63	0.45(0.650)
	Long (>5)	78	49.4	3.47	0.71		4.17	2.78	
Emergency Department (years)	Short (<5)	85	53.8	3.24	0.70	−1.83(0.069)	4.32	2.63	0.26(0.795)
	Long (>5)	73	46.2	3.44	0.72		4.21	2.79	
Current position	Staff nurse	122	77.2	3.28	0.73	−1.61(0.109)	4.26	2.65	−0.03(0.976)
	Charge/Head nurse	36	22.8	3.50	0.65		4.28	2.89	

^†^ Mean (SD); DPQ-N: Disaster Preparedness Questionnaire for Nurses; WTR: Willingness to respond.

**Table 2 ijerph-19-11812-t002:** Differences in disaster preparedness and WTR by disaster-related characteristics (*N* = 158).

Characteristics	Categories	*n*	%	DPQ-N		*t*/F(*p*)	WTR		*t*/F(*p*)
				Mean	SD		Mean	SD	
Personal disaster experience (victim or witness)	Yes	40	25.3	3.68	0.75	3.65(<0.001)	4.58	2.56	0.84(0.403)
	No	118	74.7	3.22	0.67		4.16	2.75	
Disaster experience in a disaster scene as a professional	Yes	21	13.3	3.83	0.68	3.58(<0.001)	5.57	2.48	2.42(0.017)
	No	137	86.7	3.25	0.69		4.07	2.68	
KDMAT	Yes	49	31.0	3.81	0.63	6.26(<0.001)	5.02	2.54	2.39(0.018)
	No	109	69.0	3.12	0.65		3.93	2.71	
Disaster training within a year	Yes	91	57.6	3.59	0.62	5.84(<0.001)	4.76	2.71	2.73(0.007)
	No	67	42.4	2.98	0.69		3.60	2.55	
Safety concerns during a disaster	Self	49	31.0	3.22	0.80	0.53(0.586)	4.31	2.67	0.64(0.531)
	Family	135	85.4	3.34	0.69		4.26	2.73	
	No concern	9	5.7	3.38	0.61		3.22	3.19	
Available resources during disaster	Yes	33	20.9	3.55	0.67	2.10(0.126)	4.12	2.61	0.67(0.513)
	No	83	52.5	3.26	0.75		4.49	2.76	
	Not sure	42	26.6	3.30	0.65		3.93	2.67	
Compensation for disaster participation	Yes	31	19.6	3.51	0.71	2.13(0.122)	4.19	2.89	0.26(0.772)
	No	80	50.6	3.22	0.70		4.41	2.64	
	Not sure	47	29.7	3.40	0.72		4.06	2.71	

DPQ-N: Disaster Preparedness Questionnaire for Nurses; WTR: willingness to respond; KDMAT: Korean Disaster Medical Assistance Team.

## Data Availability

The data that support the findings of this study are available from the corresponding author upon reasonable request.

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
