# Peer review of "Perceived Disaster Preparedness and Willingness to Respond among Emergency Nurses in South Korea: A Cross-Sectional Study"

_ijerph, 2022, doi:10.3390/ijerph191811812_

Round 1

Reviewer 1 Report

Thank you very much for the opportunity to review the manuscript.

 I as a reader thought that this paper had a systematic structure as a research article. Nonetheless, after reading the manuscript, I still wondered what would be unique characteristics of test results in this manuscript. Several findings in this manuscript did not look much different from those in previous studies (Or your findings were a little known in the Korean society and thus somewhat discussed in public meetings). To the token, I encourage the authors to provide the peculiar characteristics of tests results somewhere with the revised manuscript.

Author Response

Thank you for your valuable comment, and please see the attachment.

Reviewer 2 Report

General comments:

The present study is interesting research, and I wish to congratulate the team. However, I would like to make some very essential comments that require revisions or clarifications. Firstly, the study was conducted almost three years ago. How would the authors link to the current situation?

Specific comments:

Abstract:

Highlighting some crucial results with the weightage (frequency, proportion, significant level etc.) could be helpful for the readers

Introduction:

In general, it is well written. However, kindly focus on spacing, punctuation, etc. For example, the authors mentioned “health care” and other sections as “healthcare” in one line. Kindly make it uniform as per the CDC and WHO guidelines.

Methods:

Sample size – Usually, we use CDC, WHO sample size calculator, or Cochrane sample size formula. As a reviewer, I feel the sample size is small. For justification of the selected sample size, kindly provide a proper reference.

Data analysis: Authors have performed parametric tests (t-tests and ANOVA). Did the authors perform any analysis to confirm the distribution of the data? 

(normal or skewed). More explanation of the values is required)

Results:

Properly presented. However, results are related to parametric. Kindly link with the methodology

Discussion:

No comments

Author Response

Thank you for your valuable comments, and please see the attachment.

Round 2

Reviewer 1 Report

Your revision has been satisfactory to me. 

Reviewer 2 Report

Dear authors,

Thank you very much for incorporating my comments to improve the article quality.

Wish you all the best